# The influence of collection method on paleoecological datasets: In-place versus surface-collected fossil samples in the Pennsylvanian Finis Shale, Texas, USA

Frank L. Forcino ◉*, Emily S. Stafford◉

Department of Geosciences & Natural Resources, Western Carolina University, Cullowhee, North Carolina, United States of America

◉ These authors contributed equally to this work.
* Flforcino@email.wcu.edu

**Data Availability Statement:** All relevant data are within the paper and its Supporting Information files.

## Abstract

There are multiple common methods for collecting fossil material in the field for paleoecological analyses, so it is important to determine if and how different methods may affect the similarities and differences among taxonomic samples. Here, we evaluate the influence of two fossil collection field methods (stratigraphically in-place bulk-sediment versus picking up weathered-out fossils from the ground surface) on paleoecological results, using the Pennsylvanian marine invertebrate assemblages of the Finis Shale in Texas. Based on an informal review of recent paleoecology papers, we observed that the lithology of the study material and the nature of the research question correspond to choice of field collection protocols; however, collection protocols are not always clearly explained or justified in the text of the papers. For the present case study, we collected stratigraphically equivalent samples from three outcrops using both the surface pick-up and in-place bulk sediment methods. We found a difference in the abundance and composition of paleocommunities between these two collection methods. Evidence to support this includes the significant differences between samples using PERMANOVA (p < 0.001), the clear separation in ordination space of samples clustered by sampling method, the significantly higher richness in the surface samples (p < 0.001), and the considerable variation in relative abundances of various taxa and taxonomic groups. Richness and evenness were higher among the surface-collected samples, possibly due to collector bias, weathering artifacts, or spatial and temporal variability. Paleontologists strive to do the best science possible with the material available. Often, paleoecological research methods are limited by time, funding, or the nature of the material. In such cases, we recommend examining both collection methods, even if for only a fraction of the sampling. If only one method is possible, we recommend the use in-place, bulk-collected samples.

**Funding:** The author received no specific funding for this work.

**Competing interests:** The authors have declared that no competing interests exist.

## Introduction

The methods employed in paleontological research are ideally determined by the goals of the study [1–4], but can also be influenced by locality and outcrop availability [5,6], lithology [7], funding or time limitations, or even researchers' specific expertise or traditions within the discipline. Previous work has sought to determine the most appropriate field methods [3,4,8,9], lab methods [2,8], and statistical techniques [4,10] to use in paleoecological research. These studies all have the ultimate goal of helping researchers maximize both the quantity and quality of information from the available fossil data. Paleontologists, like all scientists, want to get the right answers; thus it is important to continually evaluate the methods employed for each study. In this paper, our goal is to understand the influence of different fossil collection field methods on paleoecological results, by conducting a informal review of recent paleontology literature and a case study using the Pennsylvanian marine invertebrate assemblages of the Finis Shale of Texas.

To understand ecological conditions of the past, paleontologists often compile faunal presence and abundance of fossil assemblages [1,11,12]. Irrespective of the specific goal of a paleoecological study, the taxonomic identifications and abundance counts used to determine fossil community distributions can be done in the field [e.g., 3,5,13–15] or fossils can be collected and brought back to the lab [e.g., 16–20]. When collecting fossils to bring back to the lab, two different fossil collection methods are typically employed: (1) bulk collection of stratigraphically in-place fossil-containing sediment or rock, and (2) surface collection of loose, individual fossils that have weathered out of an outcrop.

Although in-place, bulk sampling allows more stratigraphic and spatial control, there are occasions when time, money, lithology, or outcrop availability may preclude bulk sampling. Certain types of studies may use surface collection to attain appropriately large sample sizes (e.g., predation assessments [21]). Each method requires varying levels of effort and expertise. Bulk collection requires time-consuming disaggregation before the fossils can be identified and counted.

Here, we compare surface-collected samples to stratigraphically corresponding in-place, bulk-collected samples to compare the resultant fossil assemblages. If the composition and abundance of taxa—paleocommunities—are the same between the two collection methods, then researchers can be confident that collecting from the surface will provide an accurate recreation of past ecosystems. If the paleocommunities differ between the two collection methods, then researchers must consider this when planning field collections. If researchers use inappropriate collection methods due to financial or time constraints, and the resulting data are not scientifically meaningful, the time and money have been wasted.

### Review of field-based collection methods in the literature

Prior to conducting the present case study, we did a review of recent paleoecological literature to see the prevalence of different collection methods employed by paleontologists. The goal of this review is to compile the prevalence of different field-based collection methods and determine why particular methods are chosen for particular types of research.

We examined articles from *Palaios*, *Paleobiology* (*Pb*), and *Palaeogeography*, *Palaeoecology*, *Palaeoclimatology* (*P3*). In *Palaios* and *Pb*, we examined research articles in all volumes from 2014 to 2018. For *P3*, we covered only paleontology research articles from all 2018 volumes to obtain data from approximately the same number of articles from each journal, with a target of approximately 200 total articles each. From this point, the articles were culled to select articles that 1) studied aquatic macroinvertebrates (i.e., not terrestrial organisms, vertebrates, plants, microbes, or microfossils); and 2) were based on fossil material (i.e., not death

assemblages, living organisms, or simulations) that 3) was field-collected by the authors for the study at hand (i.e., not museum collections, databases, or literature reviews).

These articles' topics were categorized as community paleoecology, biotic interactions, taphonomy/preservation, and "other" (e.g., biostratigraphy, paleoenvironmental interpretation). The fossil collection methods were categorized as "pickup" (surface collected from the ground or outcrop), slab/section (methodically counted on hard surfaces or through stratigraphic sections; e.g., point-counting), or bulk sediment (from volumes of disaggregated sediment or rock).

We tabulated 703 total articles, 93 of which used field-collected, fossil aquatic macroinvertebrate material. Slab/section was the most common method (44 articles), followed by pickup (40 articles), then bulk-sediment sampling (23 articles); some articles used more than one method (and thus were counted for both methods). For 28 of these 93 articles, the collection method was not clearly stated in the article. When these "uncertain" articles were assigned methods (based on other details from the articles, such as the descriptions of the sampling localities or the nature of the data), the distribution was similar, though less strongly dominated by slab/section sampling.

We then compared collection methods with article topics (Fig 1A) and with lithology of the study material (Fig 1B). Among community paleoecology articles, taphonomy/preservation articles, and articles on "other" topics, slab/section was the most commonly used method (11 of 22 articles, 13 of 25, and 18 of 32, respectively). In contrast, in articles on biotic interactions, slab/section sampling was the least used (2 out of 14) and bulk-sediment and pickup were roughly equal (Fig 1A).

For lithology, slab/section sampling was the most common method among articles that featured only carbonate material (20 of 30 articles), and bulk-sediment was the least-used (5 articles) (Fig 1B). In articles featuring only siliciclastic or volcaniclastic material, pickup was the most-used method (15 of 34), but the methods were more evenly represented. In articles where both carbonate and clastic material were used, pickup was the most-used method, and bulk sediment was the least (13 versus 4 out of 24 articles) (Fig 1B).

One reason slab/section sampling prevails may be that carbonate rocks usually do not allow bulk sediment disaggregation and are less likely to produce abundant weathered fossils for pickup. Point-counting from the surfaces of slabs, or analysis of stratigraphically-collected thin-sections [7], may be the only practical methods for certain localities. This is born out in the data (Fig 1B). Slab and section sampling may also be more useful for colonial or framework-building taxa (i.e., bryozoans, corals, and sponges) that cannot be easily collected as individual fossil specimens. A circumstance where pickup (surface) sampling may be preferred is in cases where the fossils occur as films or impressions (e.g., studies on exceptional preservation), where the fossils cannot be separated from the sediment. Studies where full paleocommunity data are not needed, such as taxonomic descriptions of particular specimens, may also require only a few surface-collected specimens.

Although bulk-sediment collection have undergone the least taphonomic alteration and may offer the most accurate data on ancient communities, bulk collection did not dominate any of the topic or lithologic categories. Authors generally did not justify why one method was chosen over other, potentially more-accurate methods. One goal of the present study is to determine whether such justifications are necessary.

In almost one-third of the papers included within our informal review, the collection methods were not clearly enough explained to allow us to confidently identify the collection method. If collection methods are not well-explained, it is difficult or impossible for readers to ascertain the validity of the methods, and by extension, the validity of the results. Additionally, researchers interested in recreating similar analyses would be unable to accomplish this goal.

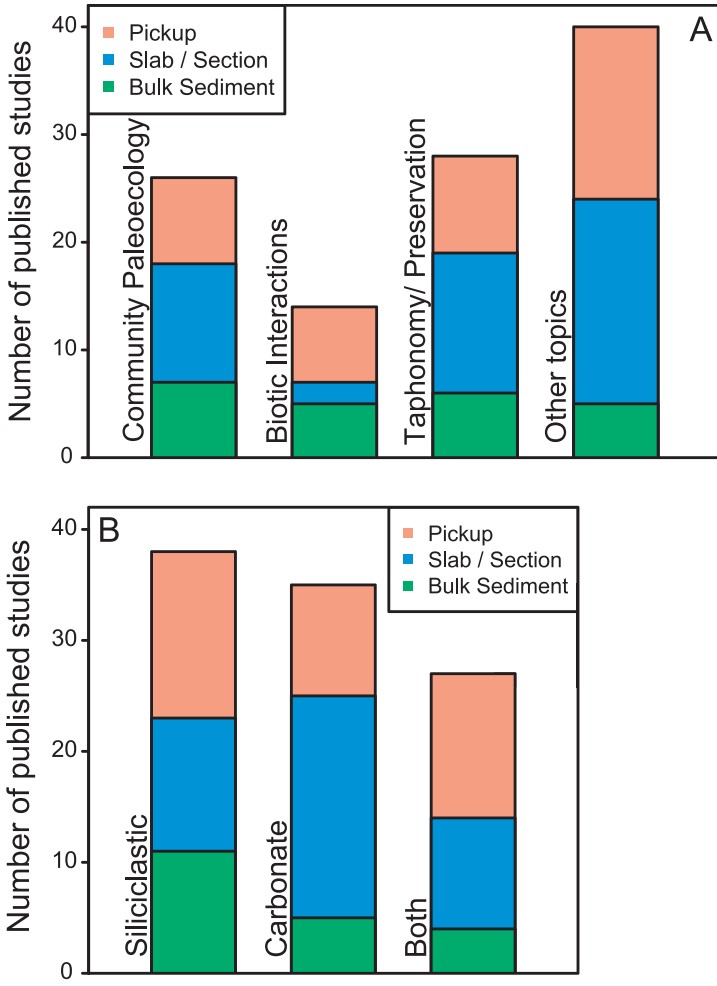

**Fig 1. Informal review of fossil collection methods in the literature.** A. Fossil collection methods by article topic. B. Fossil collection methods by lithology of the study material. Note, the bar height represents the total number of methods used, not the total number of papers, because single papers sometimes used multiples methods.

The use of the pickup and bulk-sediment sampling methods in recent published studies, sometimes without explanation or justification for the method used, demonstrates the importance of directly comparing these two methods. If they produce different results, this confirms the importance of carefully considering the choice of collection method before sampling and of clearly explaining the sampling strategy in publications.

## Methods

### Geologic background

For the present case study, we sampled three different outcrops of the Pennsylvanian (Virgilian) Finis Shale of Jacksboro, Texas. The Finis Shale is an appropriate locality for this type of methodological comparison because it contains abundant and diverse fossil assemblages [3,8]. Fossils are abundant on the ground surface and within the in-place rock, with a typical sample size of 200 fossils per 4 liters of rock [3,8,10]. Furthermore, the Finis Shale provides enough stratigraphic and spatial extent for those variables to be included in the analyses (Fig 2).

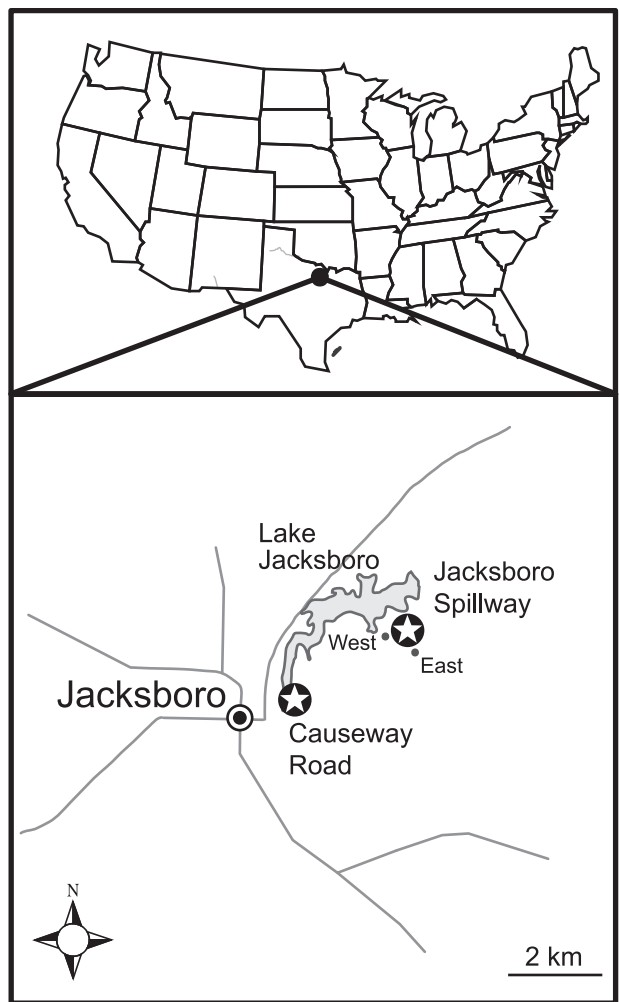

**Fig 2. Locality maps.** (A) Location of Jacksboro, TX. (B) Location of the three Finis Shale outcrops sampled: the Spillway West (33˚ 14.3' N, 98˚ 7.3' W), Spillway East (33˚ 14.3' N, 98˚ 7.3' W), and Causeway Road (33˚ 13.3' N, 98˚ 8.8' W) outcrops.

The Finis Shale was deposited along the paleoequator on the Eastern Shelf of the Midland Basin in what is now Texas, USA (Fig 2). Deposition consisted of fine-grained terrigenous sediment that settled out of suspension in a calm, low-energy environment over a span of approximately 2 million years [22,23]. Sediment grain size is consistent (<0.1 mm) both laterally (>10 km) and vertically (>5 m). The strata are essentially flat-lying over great distances, with little to no dip at the Jacksboro Spillway or Causeway Road localities. It was deposited in a low energy setting as evidenced by high carbonate mud content [24].

## Data collection

Samples were collected from three outcrops at two locations of the Finis Shale in Jacksboro, TX. At the Lake Jacksboro Spillway locality (33˚ 14.3' N, 98˚ 7.3' W) samples were collected from 3-m stratigraphic sections at both the east and the west ends of the outcrop (Spillway West and Spillway East). No permits were required for the described study, which complied with all relevant regulations. A 1-m of stratigraphic section was sampled at the Causeway Road

(33˚ 13.3' N, 98˚ 8.8' W) locality. These three outcrops were selected because of their accessible and consistent exposure, their abundance of surface and in-place fossils, and the fact that they were overlain by only a small layer of the Jacksboro Limestone, minimizing the presence of fossils weathered from the limestone above. See Forcino et al. [8] and Forcino et al. [39] for more detail about the Finis Shale outcrops at which we samples.

At each of the three sampled outcrops, two sets of samples were collected at equivalent stratigraphic positions: in-place bulk-sediment samples and surface-collected samples. For the in-place samples, we collected 4 L of bulk sediment every 1 m, beginning 1 m below the Jacksboro Limestone, and working downward stratigraphically. After clearing the outcrop surface of weathered material and vegetation, 4-L of shale was removed with a trowel and placed in gallon-size bags. The 4-L volume was chosen because in previous research this volume yielded a sample size of ~200 individual fossils [4,8,10]. We aimed for 200 individuals per sample to ensure we had ample fossils for comparison with the surface collections. The 1-m stratigraphic interval was chosen to correspond with the 1-m vertical coverage of each surface sampled area.

For the surface-collected samples, 1-m vertical extents were marked off, adjacent to and corresponding to the vertical interval between two in-place bulk samples. Two to four people (at least one faculty member and one student; for consistency, one of the faculty members surface-collected for every sampled area) exhaustively hand-picked fossil material in a ~0.25 m$^2$ area before moving to an adjacent portion of the surface, until approximately 1 L of material was collected, taking between 30 minutes to 1 hour. The 1-L sample volume was chosen to acquire at least 200 specimens, corresponding with the in-place sample sizes.

In-place, bulk sediment samples were washed and wet-sieved, then dry-sieved through a 2 mm sieve to exclude smaller specimens. Surface-collected fossils were also washed and sieved to retain specimens larger than 2 mm. The 2-mm size cutoff was chosen to enable easy identification of the fossils and to keep the sample sizes manageable (smaller specimens would greatly increase sample size, thus increasing the time taken to identify and count all specimens). Brachiopod, mollusk, coral, and foraminifera specimens were identified to the genus level. Bryozoans and crinoids were identified and included in the analysis based on morphology (e.g., stick or fenestrate bryozoans). All other higher taxa included in the analysis were monospecific.

Abundance counts were completed using the minimum number of individuals (MNI) technique [25]. This technique gives a count of one to articulated brachiopod or bivalve specimens and a count of one to the most abundant valve (pedicle/brachial and left/right) of disarticulated specimens. For brachiopods, the cardinal process (on brachial valves) or the umbo or pedicle foramen (on pedicle valves) had to be present for the individual to be counted. For bivalves, the umbo had to be present. For gastropod and cephalopod taxa, a landmark was chosen to prevent double counting. For most gastropods, the shell spire (minus the protoconch) was chosen as the countable landmark, as long as enough spire whorls were present to identify the taxon (thus body whorls without the spire were not counted). For mollusks in which the early whorls were enveloped within later whorls (e.g., bellerophontoid gastropods and coiled cephalopods), the central portion of the shell was counted. However, if no countable specimens of a taxon were present, then an identifiable fragment of that taxon was counted as an abundance of one (to indicate the presence of that taxon in the sample).

Colonial (e.g., bryozoans and corals) and easily disarticulated taxa (e.g., crinoids) were included as presence or absence (P/A). Including P/A data of the colonial and easily disarticulated taxa would underestimate their relative abundance and might not provide meaningful comparisons between in-place and surface samples; however, Forcino and Leighton [26] showed no differences in paleocommunity results between using an abundance count (2 cm equaling one individual) and P/A within the Finis Shale. We used P/A to keep comparisons as

simple as possible, and we also conducted a separate analysis using only brachiopod and mollusk data (because brachiopods and mollusks are more easily identifiable as fossils and are larger, they may be more likely to be picked up from the surface than bryozoans or crinoids). We also conducted analyses with and without the P/A taxa included in the analysis (see next section for details).

All fossil specimens used in this study are stored in the Stillwell Building at Western Carolina University, Cullowhee, NC. All samples were collected on public lands. No endangered or protected species or locations were involved in this study.

## Statistical analysis

Statistical analyses were conducted using R 3.4.2 and the vegan package [27,28]. Generic richness and evenness were calculated for each sample for both collection methods. Paired t-tests were conducted to test for differences in richness and evenness between the two collection methods.

We compared the samples using non-metric multidimensional scaling (NMDS) ordinations, a multivariate exploratory technique. Additionally, we tested for multivariate statistical differences using Permutation Analysis of Variance (PERMANOVA [29]). Ordination is an exploratory analysis that is effective at expressing multidimensional relationships within a few dimensions. The resulting ordination plot orders the samples along axes according to their constituent genera and distribution of abundances, with more similar samples plotting closer in ordination space [4,30,31]. We used NMDS ordination (with Bray-Curtis similarity measures) because it is frequently used in both ecology [32–34] and paleoecology [35–37], and is considered the most effective ordination technique for capturing community gradients [4,9,32,38].

A separation among samples within ordination space by collection method (along with a PERMANOVA p-value less than 0.05) would indicate that the two sampling methods drive greater differences among the paleocommunity metrics than collector, stratigraphic position, or spatial location. If samples are grouped in ordination space by collector, stratigraphic position, or location, this would indicate that one of these factors drives differences in the abundance and distribution of genera.

Three datasets were analyzed: (1) abundance counts of mollusks, brachiopods, solitary corals, fusulinid foraminifera, with presence/absence of colonial and easily disarticulated taxa (i.e., bryozoans and crinoids); (2) presence/absence of all taxa; and (3) abundance counts of mollusks and brachiopods only. We analyzed the all three datasets with each collectors' samples kept separate, and then with all collectors' samples combined.

## Post hoc subsampling

Because sample size differences between the in-place and surface-collected samples could drive differences in apparent paleocommunities, we subsampled 25% of each surface sample from the Spillway West outcrop, and compared the subsamples to the 100% surface sample and to the equivalent in-place sample. For consistency, our subsampling analysis included only samples collected by the one collector who surface-sampled all locations. To produce surface-subsamples of a similar size to the in-place samples, we randomly mixed, then twice halved the surface samples, producing two 25% subsamples each. Although using a splitter may have produced a more ideal subsample, the variability in specimen size and the fragility of many specimens made using a splitter impractical. The subsampled fossils were then sorted, identified to the genus level, and counted. NMDS ordination was used to compare the relative abundances of taxa between each sample and subsample.

## Results

The total number of fossil individuals in the in-place, bulk-sediment samples was 1,836 from seven samples, and the total number of fossil individuals in the surface samples was 8,030 among seven sample areas. The median sample size of the in-place samples was 222 individuals, and 1176 for the surface samples (Table 1). The majority of fossils were brachiopods or mollusks. Minor components of each sample included bryozoans, crinoids, fusulinid foraminifera, sponges, corals, echinoids, conulariids, trilobites, and shark teeth.

The generic richness of the in-place bulk samples ranged from 18 to 34, and the richness of the surface samples ranged from 44 to 57 (Table 2). The evenness of the in-place samples ranged from 0.54 to 0.75, and the evenness of the surface samples ranged from 0.48 to 0.77 (Table 2). Spatial and stratigraphic trends in richness and evenness were different among the in-place and surface samples. Using a paired t-test, there was a significant difference between the in-place and surface-collected sample richness ($p < 0.001$), but not between the evenness ($p = 0.35$; Table 2).

NMDS ordination of all three datasets, with collectors both combined and separated, produced samples plotting in space clustered by sampling method (Figs 3 and 4). Using PERMANOVA, all six comparisons of in-place to surface samples were significantly different ($p < 0.001$ to $p = 0.003$; Table 3).

The ordination and PERMANOVA results are reflected in the differences in relative percentage of taxonomic compositions and distributions between the in-place and surface samples. In six of the seven samples (all except for Spillway West Sample 3), there was a lower relative abundance of brachiopods and a greater relative abundance of gastropods in the

**Table 1. Sample size comparisons between in-place and surface-collected samples.** Numbers in parentheses are the median sample sizes of the collector who collected from all seven surfaces. Subsampling was only necessary for surface samples, so there are no subsampling results listed for in-place samples.

| | | *In-place* | *Surface* |
|---|---|---|---|
| 100% Samples (all) | Median sample size | 222 | 1176 (778) |
| | Range | 153 to 400 | 687 to 1419 |
| 100% Samples (brachiopods and mollusks only) | Median sample size | 218 | 1079 (768) |
| | Range | 149 to 393 | 630 to 1229 |
| 25% Subsamples (all) | Median sample size | | 239 |
| | Range | | 142 to 289 |
| 25% Subsamples (brachiopods and mollusks only) | Median sample size | | 209 |
| | Range | | 136 to 281 |

**Table 2. Diversity metrics.** Richness and evenness for the brachiopod and mollusk dataset.

| | Richness | | Evenness | |
|---|---|---|---|---|
| | *In Place* | *Surface* | *In Place* | *Surface* |
| Spillway W 3 | 34 | 46 | 0.66 | 0.48 |
| Spillway W 2 | 26 | 49 | 0.59 | 0.76 |
| Spillway W 1 | 20 | 57 | 0.54 | 0.76 |
| Spillway E 3 | 19 | 53 | 0.75 | 0.59 |
| Spillway E 2 | 30 | 54 | 0.72 | 0.74 |
| Spillway E 1 | 31 | 46 | 0.56 | 0.77 |
| Causeway | 18 | 44 | 0.54 | 0.69 |
| *Paired T-test* | $p < 0.001$ | | $p = 0.35$ | |

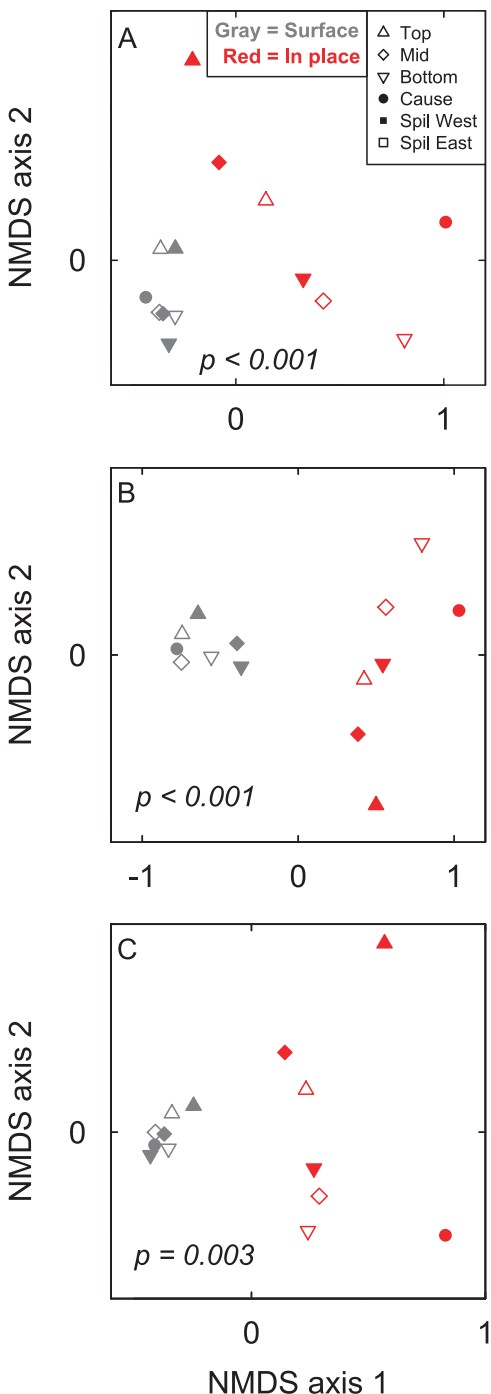

**Fig 3. NMDS ordination of all samples.** Non-metric multidimensional scaling (NMDS) ordination of all samples with all collector's surface samples combined for (A) all taxa with abundance counts, (B) all taxa using presence/ absence, and (C) mollusks and brachiopods with abundance counts. Spillway East outcrop samples are represented by open points, Spillway West outcrop samples are represented by closed triangles and diamonds, and Causeway Road outcrop samples are represented closed circles. P-values result from the PERMANOVA testing for differences between the two sampling methods.

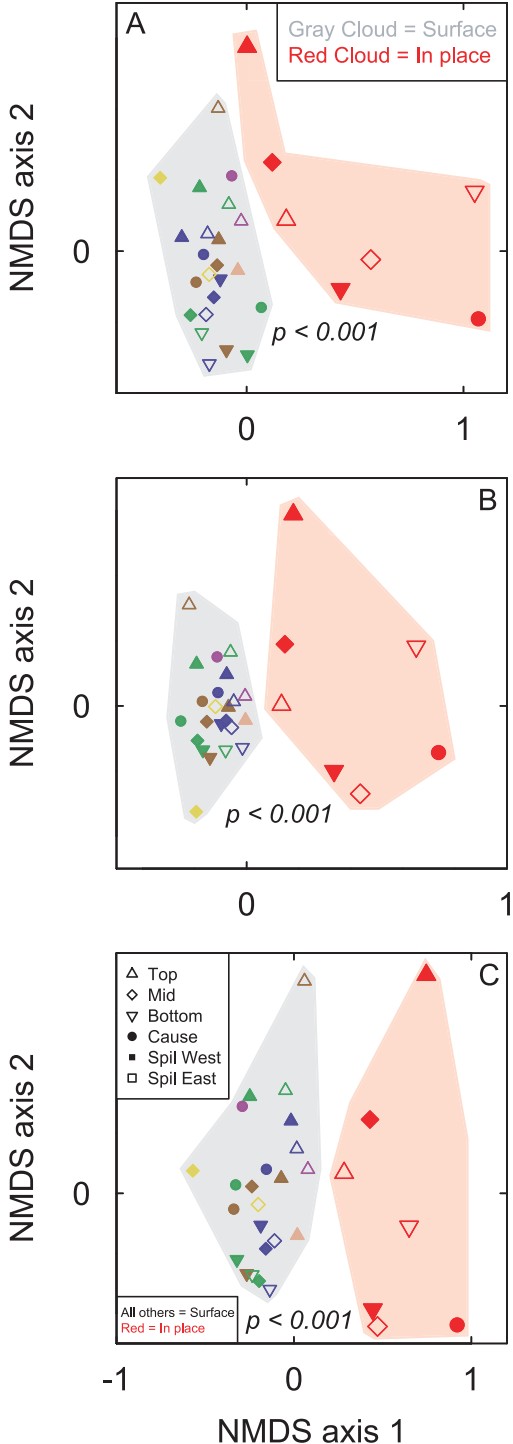

**Fig 4. NMDS ordination of samples by collector.** Non-metric multidimensional scaling (NMDS) ordination of all samples, with all surface samples identified by individual collector (as identified by the color of the smaller, non-red points), for (A) all taxa with abundance counts, (B) all taxa using presence/absence, and (C) mollusks and brachiopods with abundance counts. Spillway East outcrop samples are represented by open points, Spillway West outcrop samples are represented by closed triangles and diamonds, and Causeway Road outcrop samples are represented closed circles. Larger red points are in-place samples. Smaller points of all other colors are surface samples. Each of the six colors (other than red) represents a different surface collector. The gray and red shading are meant to further distinguish the in-place and surface collected sample points. P-values result from the PERMANOVA testing for differences between the two sampling methods.

**Table 3. PERMANOVA results.**

|  | p-value |
|---|---|
| All taxa by collector | < 0.001 |
| All taxa combined | < 0.001 |
| Brachs / Mollusks by collector | < 0.001 |
| Brachs / Mollusks combined | 0.003 |
| P/A by collector | < 0.001 |
| P/A combined | < 0.001 |

surface samples compare to the in-place samples (Fig 5). For example, in the Spillway East Sample 1 in-place, there were 85% brachiopods and 8% gastropods; in the equivalent surface sample, there were 50% brachiopods and 33% gastropods (Fig 5). In six of the seven samples (all except for Spillway East Sample 1), there was a greater relative abundance of bivalves in the in-place samples than in the surface samples (Fig 5). For example, in Spillway West Sample 3 there were 16% bivalves in the in-place sample, but 3% bivalves in the surface sample.

Because colonial and easily disarticulated taxa were recorded as presence/absence data, the following summaries of abundance and sample size totals are for brachiopods and mollusks only. The relative abundances of the three most abundant taxa (*Crurithyris*, *Neochonetes*, *Rhipodomella*) differed between the in-place and surface samples (Fig 6). For example, in the Spillway West Sample 1, the in-place sample was 63% *Crurithyris*, but 12% in the surface sample. In the Spillway West in-place Sample 3, there were 21% *Rhipodomella*, but 61% in the surface sample (Fig 6). Another abundant brachiopod taxon, *Hustedia*, made of 25% of the in-place Spillway East Sample 3, versus only 3% in the surface sample. In Spillway West Sample 2, the productide brachiopod clade made up 0% in-place versus 16% on the surface. Furthermore, in the Spillway West Sample 1, the bivalve *Astartella* was 7% in-place versus 4% on the surface, and the gastropod *Straporallus* was 1% in-place versus 6% on the surface.

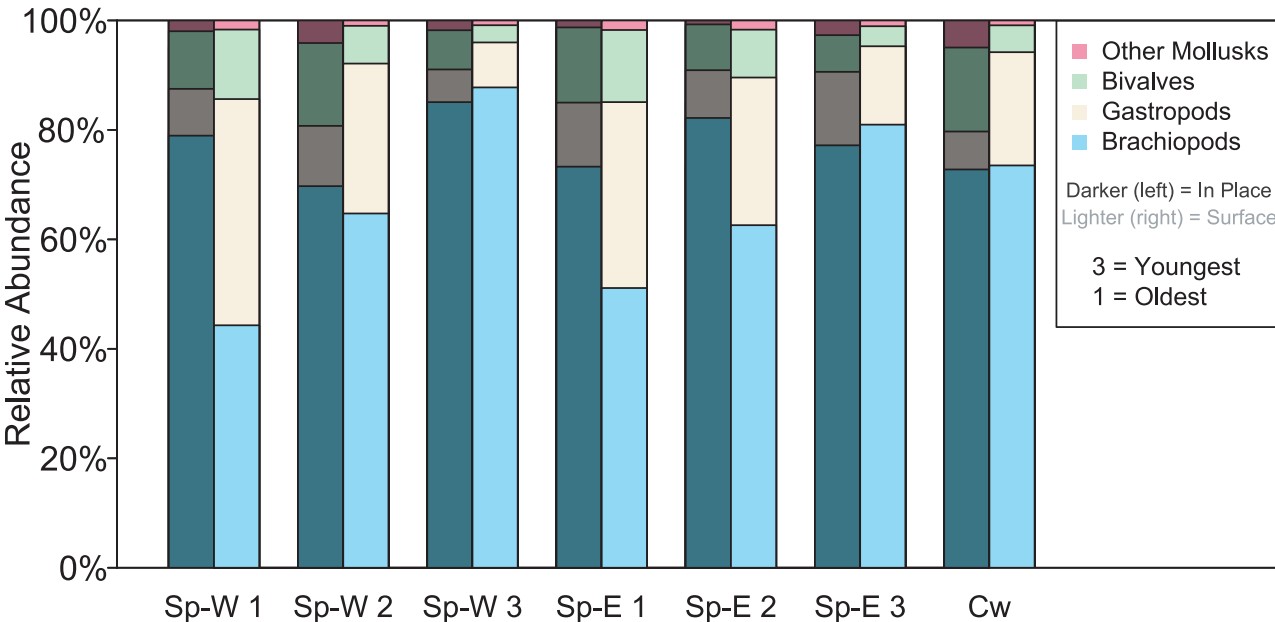

**Fig 5. Relative abundance: In-place versus surface.** Relative abundance of major taxonomic groups compared between the in-place and surface samples. SpE = Spillway East outcrop, SpW = Spillway West outcrop, CW = Causeway Road outcrop. The lower the number, the lower stratigraphic position.

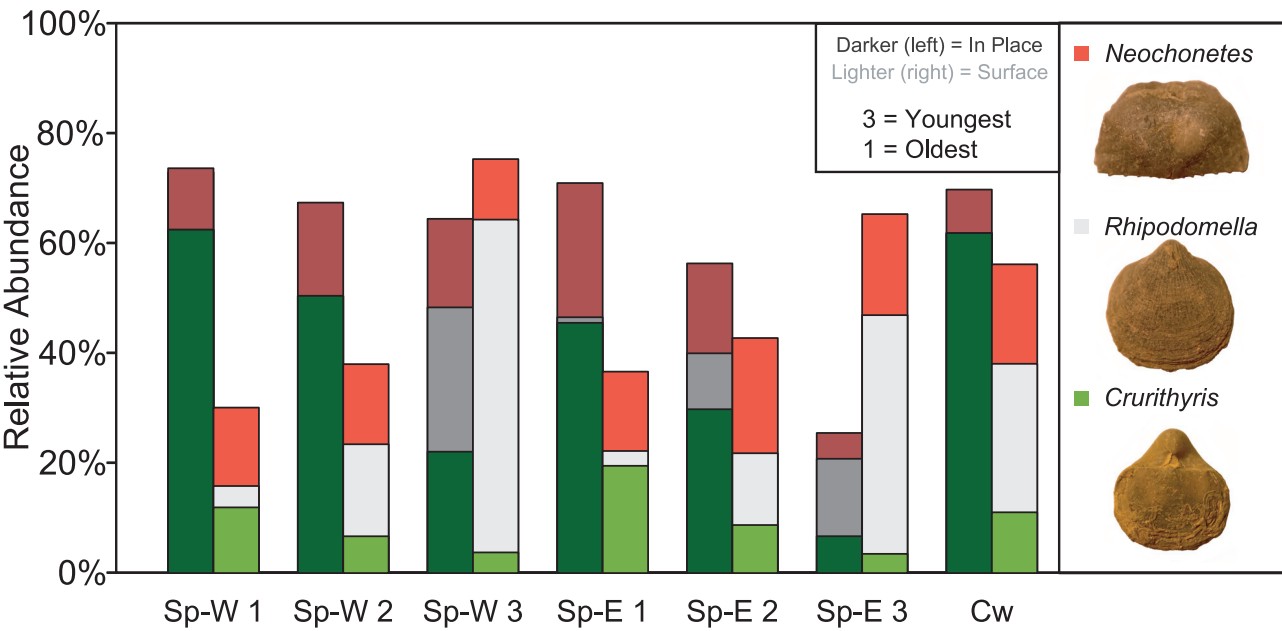

**Fig 6. Relative abundance of select brachiopod genera.** Relative abundance of the three most abundant genera (all brachiopods) across all samples compared between the in-place and surface samples. SpE = Spillway East outcrop, SpW = Spillway West outcrop, CW = Causeway Road outcrop. The lower the number, the lower stratigraphic position.

## Post-hoc subsample comparison

The median sample size for the 25% surface-subsamples was 201, similar to the in-place median sample size of 222 (Table 1). The median richness in the surface-subsamples was 25, closer to the in-place samples (richness = 25) than the 100% surface samples (richness = 45; Tables 2 and 4). Conversely, the median evenness for the surface-subsamples was 0.80, closer to the 100% surface samples (evenness = 0.74) than the in-place samples (evenness = 0.58). In NMDS ordination, surface samples separated from in-place samples, regardless of sample size (Fig 7). The relative abundances of brachiopods and mollusks were more similar between the 25% and 100% surface samples than the in-place samples (Fig 8).

## Discussion

The fossil taxonomic composition, distribution, and abundance of the seven in-place, bulk-collected samples differ from those of the stratigraphically equivalent surface-collected samples (Figs 3–6). Evidence to support this includes the significant differences between samples using PERMANOVA (p < 0.001; Table 3), the clear separation in ordination space of samples clustered by sampling method (Figs 3 and 4), the significantly higher richness in the surface samples (p < 0.001; Table 2), and the considerable variation in relative abundances of various taxa and taxonomic groups (Figs 5 and 6). Furthermore, the spatial and stratigraphic variations in community composition are disparate between the in-place and surface samples. This indicates that relative variation in the ecosystems and the parameters that accompany those communities would be interpreted as different based off the two different collection methods. Because the closest community to that of the once living animals is the community that has undergone the fewest taphonomic events (in this case weathering out of an outcrop), the in-place, bulk-collected sample will likely lead to a more accurate paleocommunity representation

**Table 4. Subsample diversity metrics.** Richness and evenness for the in-place samples, 100% surface samples, and 25% subsamples for the brachiopod and mollusk dataset. Surface sample values are for only the one collector who sampled from all surfaces. The values in parentheses are those for all collectors' samples combined, and are the same as those in Table 2.

| | Richness | | | | Evenness | | | |
|---|---|---|---|---|---|---|---|---|
| | *In- Place* | *100% Surface* | *25% subsample* | *25% subsample* | *In- Place* | *100% Surface* | *25% subsample* | *25% subsample* |
| Spillway W 3 | 34 | 37 (46) | 22 | 19 | 0.66 | 0.56 (0.48) | 0.65 | 0.66 |
| Spillway W 2 | 26 | 48 (49) | 31 | 35 | 0.59 | 0.76 (0.76) | 0.82 | 0.82 |
| Spillway W 1 | 20 | 50 (57) | 26 | 28 | 0.54 | 0.75 (0.76) | 0.78 | 0.84 |

than using surface-collected samples when conducting an ecological assessment of fossil communities. Depending on the goals and type of paleontological research (i.e., something other than paleocommunity ecology), surface collecting should be clearly justified when it is used.

Because results for all six dataset comparisons demonstrate the same patterns and reveal similar statistical results from the PERMANOVA, we will discuss the possible reasons for our results and make interpretations based on the mollusk and brachiopod (all collectors combined) dataset, to avoid repetition throughout the discussion. We chose the mollusk and brachiopod dataset in particular because it had the highest PERMANOVA p-value (p = 0.003) of the six dataset comparisons and thus is the most conservative for testing the null hypothesis (that there were no differences between the multivariate paleocommunity results between collecting in-place and surface samples).

## Sample size

The median in-place sample size was 222, and the median surface-collected sample size was 1176 (Table 1). To determine if the community differences found between the two different methods was driven by the larger sample size in the surface samples, we compared 25% subsamples to the complete surface samples as well as to the in-place samples (Table 4; Figs 7 and 8). Sample size and generic richness in the 25% surface subsamples were closer to the in-place samples than to the 100% surface samples (Table 4). On the other hand, the diversity metrics indicate that although the 25% subsamples may not contain all the rare taxa found in the 100%

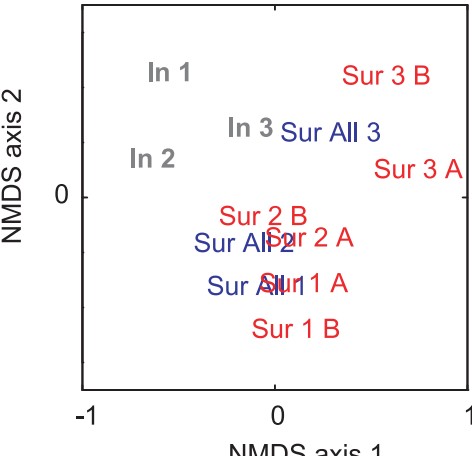

**Fig 7. NMDS ordination of Spillway West samples.** Non-metric multidimensional scaling (NMDS) ordination of the three Spillway West samples separated into in-place (In), 100% surface sample (Sur All), and the two 25% surface subsamples (A and B). The lower the number, the lower stratigraphic position.

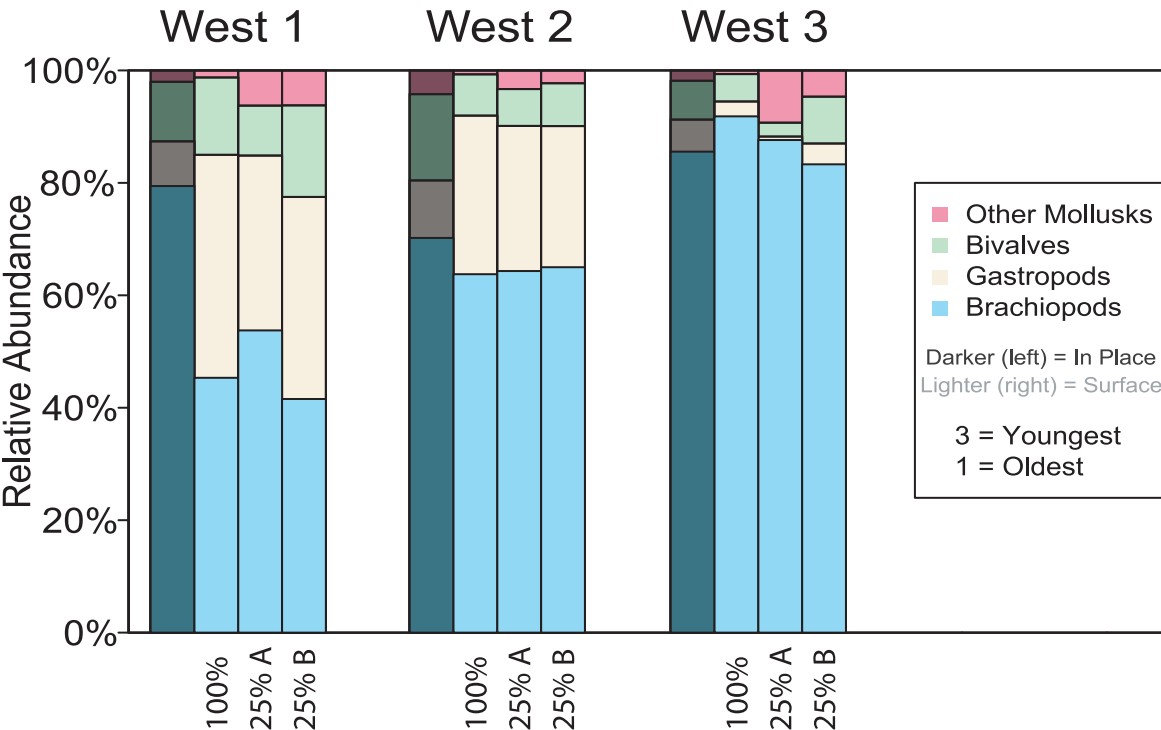

**Fig 8. Relative abundance of major taxonomic groups.** Relative abundance of major taxonomic groups in the three Spillway West samples, compared between the in-place sample, the 100% surface sample, and the two 25% subsamples (A and B). The lower the number, the lower stratigraphic position.

surface samples, the abundance distributions are preserved. In NMDS ordination, the surface samples separate from the in-place samples, regardless of sample size (Fig 7). The relative abundances of brachiopods and mollusks are more similar between the 25% and 100% surface samples than the in-place samples (Fig 8). Overall, the 25% subsamples are not similar enough to the in-place samples to suggest that sample size drove the observed differences between the 100% surface and in-place samples. Thus, we will be comparing the 100% surface samples to the in-place samples for the remainder of the discussion.

### Richness

The surface collected samples had higher richness than the equivalent in-place sample (Table 2). Among the surface samples, higher sample size coincided with generic richness (Tables 1 and 2). However, this is not the case for the in-place samples. Forcino et al. [8], Forcino [10], and Forcino et al. [39] found that the richness of the Finis Shale in-place samples would not be significantly greater even if up to four times the volume of bulk sediment were collected. Additionally, when the surface samples were randomly subsampled to 25% of their original sample size, the richness was still higher than the in-place samples (Table 4). Thus, sample size did not solely influence the difference in richness between the two collection methods.

The observed richness in the surface samples was greater than that of the in-place sample for reasons that could influence the paleoecological interpretations of a study. Possible explanations for the greater richness in the surface samples include (a) fossils weathered from stratigraphically higher rocks are adding to the richness at lower stratigraphic levels and (b) a larger spatial and temporal stratigraphic extent was covered when we collected the surface samples. If

(a) were the case, richness would be predicted to decrease up-section for the surface samples because there would be less mixing toward the top of the outcrop. This predicted pattern occurs at the Spillway West outcrop, but not at the Spillway East outcrop (Table 2). Although there is no consistent pattern of decreasing richness up-section, it is likely that weathering is adding to the richness of the lower surface samples. Evidence for this mixing is that fusulinids are found only in the surface samples. No fusulinids were found in any Finis Shale bulk sediment samples in any previous study conducted by the authors [3,8,40]. There is also a clear preservational difference in the fusulinids that is consistent with fossil material from the overlying Jacksboro Limestone. The presence of sponges, abundant in the Jacksboro Limestone but not documented in the Finis Shale (neither in previous studies [41,42] nor in any of the in-place samples in the present study) is additional evidence of stratigraphic mixing due to weathering. Therefore, we are confident that fossil material is weathering down from the exposure, albeit limited, of Jacksboro Limestone onto the surface of the Finis Shale. From this fact alone, a community analysis of the Finis Shale based on surface-collected specimens is not an accurate reflection of the original ecosystem at the time the Finis Shale.

The greater spatial area covered by the surface samples also could have driven the greater richness. The in-place bulk sediment collections extended 10 cm stratigraphically and 10–20 cm laterally. The outcrop surface area that was covered during surface collection spanned from 5 to 10 meters. If richness varied spatially in Finis Shale, a larger area of collection could increase total richness in a sample. Based on comparisons of richness and taxonomic composition between the Spillway West and East outcrops found in the present study (Table 2) and found in Forcino et al. [3], spatial variation does not account for higher richness in the surface samples.

Because of the greater surface area covered to gather the surface samples, it could be that the difference in richness, and possibly the differences in taxonomic composition and abundance, could be a result of the greater spatial area covered. We were able to take into account this possibility by collecting at two spots at the same locality, the east and west outcrops at the Jacksboro Spillway, which are about 200 meters apart. The assemblages from the in-place samples from two outcrops were much more similar to one another than to the surface samples of either outcrop, suggesting that the in-place assemblages do not vary considerably spatially. Even the Causeway outcrop samples, 200 meters away from the Spillway outcrops, grouped according to collection method.

### Evenness and collector bias

There was no consistent difference in evenness between the in-place and the surface samples (Table 2). There was a greater evenness in the lower stratigraphic surface samples compared to the lower stratigraphic in-place, which coincides with relatively fewer rare taxa in the surface samples. These surface samples have more evenly distributed relative abundance of taxa compared to the in-place samples that are dominated by a few taxa (Figs 5 and 6). For example, the in-place Spillway West 1 sample is dominated by *Crurithyris* (Fig 6), with less abundant *Neochonetes* and proportionally fewer other taxa. The equivalent surface sample has more even relative abundances of *Crurithyris* and *Neochonetes*, which themselves make up a lesser proportion of the overall sample. Other common taxa, such as the brachiopod *Hustedia*, the bivalves *Astartella* and *Palaeoneilo*, and the gastropods *Glabrocingulum* and *Straparollus*, do occur in the in-place samples, but have lower abundances than in the surface samples.

One possible explanation for the paucity of rare taxa in the surface samples is that when picking up fossils, humans' pattern recognition may lead one to unconsciously pick up objects of similar types. If the fossils are mixed with rocks, sediment, and other debris, collectors may leave behind fossils that are different from the preconceived pattern.

On the other hand, a concerted effort by a collector to pick up an unbiased sample could lead to the collector unconsciously avoiding the common taxa in favor of picking up novel taxa. The ideally, exhaustive collection should eliminate this, but unconscious biases can be difficult to overcome.

Another factor that could lead to the differences in taxonomic distribution, specifically fewer rare taxa collected from the surface, is fossil size. Smaller specimens are more difficult to see and therefore to collect in surface samples. If there are differences in community composition between the size fractions (as demonstrated by Pedigo et al. [43] in a Plio-Pleistocene assemblage from Florida), then rare taxa could be missed. Future research is currently underway to explore how the small size communities (< 2 mm) compare to the larger size communities (> 2 mm) and to the surface collected sample communities.

Our present data suggest that unconscious collector biases can occur when collecting surface samples, even when care is taken by experienced paleontologists to ensure unbiased, exhaustive collecting. This flaw in surface collecting could lead to skewed results in paleocommunity studies, specifically studies in which taxonomic distribution of fossils is the primary focus.

## Preservation potential

Differences in taxonomic composition between the in-place and surface samples may be related to fossil characteristics that influence preservation potential. Characteristics such as shell thickness, mineralogy, size, and morphology affect how long a fossil lasts after it has been weathered out of the rock. If thinner and smaller fossils were more likely to be destroyed during weathering, then thinner and smaller taxa would be less abundant in the surface samples. An example of a small, thin shelled taxon that is much more abundant in-place is *Crurithyris*, the most abundant taxon in the in-place samples, ranging from 9% to 63%. However, *Crurithyris* makes up only 3% to 22% of surface samples (Fig 6). Further, the abundances of many gastropod taxa are higher in the surface samples (Fig 5). The gastropods of the Finish Shale tend to have thicker shells and are of a larger size than the most abundant brachiopod taxa. A preferential loss of small, thin-shelled taxa in surface samples would produce misleading paleocommunity results. Future work could go beyond the scope of this study to determine how thickness, size, and morphology affect the likelihood of destruction or loss in weathered surface samples.

## Conclusion

There is a strong difference in the taxonomic composition and abundance distributions between in-place and surface collected fossil samples, as evidenced by the significant differences between samples using PERMANOVA ($p < 0.001$), the clear separation in ordination space of samples clustered by sampling method, the significantly higher richness in the surface samples ($p < 0.001$), and the considerable variation in relative abundances of various taxa and taxonomic groups (Tables 2 and 3; Figs 3–6). Because surface-collected fossils may be subject to stratigraphic mixing, collector bias, and taphonomic destruction during weathering, the in-place, bulk-collected samples will be more representative of the original, once-living community. Depending on the goals and type of paleontological research, surface collecting should be clearly justified when it is used. In particular, paleocommunity studies that explore taxonomic distribution and abundance, particularly those that use multivariate analytical techniques, may be less accurate if samples are hand-picked from the surface.

On the other hand, in studies where certain taxa are targeted for collection and study, some of the negative effects of surface collection may be less relevant. Additionally, if the study's outcomes would not be influenced or affected by errant taxonomic additions or abundance distributions, then surface collecting would be acceptable.

This is only one case study, on one unit, from one time, from one general depositional environment. Thus, the differences found between communities using the two collection methods may not apply to all paleontological studies. Future research is warranted, particularly additional case studies, to determine if this is the rule or an exception.

Paleontologists strive to do the best science with the material available. Often, paleoecological research methods are limited by time, funding, or the nature of the material. In such cases, we recommend examining both collection methods, even if for only a fraction of the sampling. If only one method is possible, we recommend the use of in-place, bulk-collected samples to avoid the potential drawbacks of stratigraphic mixing, differential weathering, collector bias, or other factors that might alter a paleoecological result.

## Supporting information

**S1 Table. Informal review of collection methods in recent paleoecological articles in *Paleobiology*, *Palaios*, and *Palaeogeography, Palaeoecology, Palaeoclimatology*.** Only articles on fossil aquatic macroinvertebrates using field-collected material are included. "X" indicates that the category applies to the article. A question mark "?" indicates that the collection method was not explicitly stated in the text but was inferred based on other information from the article. (XLSX)

**S2 Table. Raw abundance data.** Raw abundance counts for all samples and genera used in the analyses.
(XLSX)

**S1 Fig. Some representative specimens collected in this study.** Scale bar is 1 cm. Top row, from left: *Euphemites*, *Glabrocingulum* sp., *Phymatopleura*, two crinoid columnals, *Rhipidomella*. Second row, from left: *Glabrocingulum grayvillensis*, *Mooreceras*, *Astartella concentrica*, *Neochonetes*, *Marginifera*. Third row, from left: *Lophophyllidium*, fusulinid, *Hustedia*, *Crurithyris*, *Punctospirifer*. Bottom row, from left: bryozoan colony fragment, echinoid spine fragment.
(TIF)

## Acknowledgments

We would like to thank Kenneth De Beats for reviewing this manuscript leading to a much-improved product. We would like to thank Lindsey Leighton and Chris Schneider for helping with the initial ideas for this study, as well as help in the field. We thank Leslie Montoya, Rainee Howard, Joe Brazelton, Justin Winter, Holly Hurding-Jones, Bryce Vascik, and Don Marlor for help collecting samples in the field. We also thank Austin Lloyd and Micki Recchuiti for help processing samples and with the subsampling of the Spillway West samples.

## Author Contributions

**Conceptualization:** Frank L. Forcino, Emily S. Stafford.

**Data curation:** Frank L. Forcino.

**Formal analysis:** Frank L. Forcino, Emily S. Stafford.

**Investigation:** Frank L. Forcino, Emily S. Stafford.

**Methodology:** Frank L. Forcino, Emily S. Stafford.

**Project administration:** Frank L. Forcino.

**Software:** Frank L. Forcino.

**Supervision:** Frank L. Forcino.

**Visualization:** Frank L. Forcino.

**Writing – original draft:** Frank L. Forcino.

**Writing – review & editing:** Frank L. Forcino, Emily S. Stafford.

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
