## [Decision Letter · Decision Letter 0]

22 Oct 2019

PONE-D-19-25701

The influence of collection method on paleoecological datasets: In-place versus surface-collected fossil samples in the Pennsylvanian Finis Shale, Texas, USA

PLOS ONE

Dear Dr. Forcino,

Thank you for submitting your manuscript to PLOS ONE. After careful consideration, we feel that it has merit but does not fully meet PLOS ONE’s publication criteria as it currently stands. Therefore, we invite you to submit a revised version of the manuscript that addresses the points raised during the review process.

We would appreciate receiving your revised manuscript by Dec 06 2019 11:59PM. To enhance the reproducibility of your results, we recommend that if applicable you deposit your laboratory protocols in protocols.io, where a protocol can be assigned its own identifier (DOI) such that it can be cited independently in the future. For instructions see: http://journals.plos.org/plosone/s/submission-guidelines#loc-laboratory-protocols

We look forward to receiving your revised manuscript.

Kind regards,

Mikołaj K. Zapalski, Ph. D., D. Sc.

Academic Editor

PLOS ONE

**Journal Requirements:**

2. In your manuscript, please provide additional information regarding the specimens used in your study. Ensure that you have reported specimen numbers and complete repository information, including museum name and geographic location.

For more information on PLOS ONE's requirements for paleontology and archaeology research, see """ ext-link-type="uri" xlink:type="simple">https://journals.plos.org/plosone/s/submission-guidelines#loc-paleontology-and-archaeology-research."""

**Additional Editor Comments (if provided):**

Dear Dr. Forcino,

thank you for submitting this interesting paper on fossil sampling. There are three reviews, and I would like to ask you to integrate all these remarks; the major problem seems to be with lack of smaller taxa in the samples. Also comparisons with other published data of this kind are essential (see comments by Reviewer #1, and #2 and suggested references). Check figures (see relevant comments of Reviewer #1), and please add a stratigraphic log of investigated data - this will facilitate following the text. I would also like to emphasize the need of adding a photographic plate that includes most typical taxa.

When preparing the final version of manuscript, please refer to all reviewers queries.

Yours sincerely,

Mikołaj K. Zapalski

**Comments to the Author**

1. Is the manuscript technically sound, and do the data support the conclusions?

Reviewer #1: Yes

Reviewer #2: Partly

Reviewer #3: Yes

2. Has the statistical analysis been performed appropriately and rigorously? 

Reviewer #1: Yes

Reviewer #2: Yes

Reviewer #3: Yes

3. Have the authors made all data underlying the findings in their manuscript fully available?

Reviewer #1: Yes

Reviewer #2: Yes

Reviewer #3: Yes

4. Is the manuscript presented in an intelligible fashion and written in standard English?

Reviewer #1: Yes

Reviewer #2: Yes

Reviewer #3: Yes

5. Review Comments to the Author

Reviewer #1: The paper by Forcino and Stafford “The influence of collection method on paleoecological datasets: In-place versus surface-collected fossil samples in the Pennsylvanian Finis Shale, Texas, USA” describes how different methods of collecting fossil material may affect the similarities and differences among taxonomic samples. It focuses on statistical comparison between in-place bulk sediment samples and picking up weathered-out fossils from the ground surface. The paper is based on the case study of three outcrops of the Pennsylvanian (Virgilian) Finis Shale of Jacksboro, Texas but also consider review of over 700 paleontological papers.

At first I was a little bit confused about the topic of the paper as the advantages of bulk over picked-up samples appear rather obvious, especially in paleoecological context. However, the review of abundant papers made by Authors shows that it’s not always a case and that different methods are used in paleontological fieldworks depending on the funding, material available, time etc. Because choosing the right sampling method is extremely important for the results obtained, this problem deserves to be published and appears suitable for Plos One. The paper is of broad international interest, the text is clear, well organized and written, however some general and more specific comments listed below should be considered before publication.

General comments:

Some doubts arise with regard to the fact that the smallest specimens were excluded from the studied samples (lines 209-219), especially that in lines 509-513 Authors show how important they are for interpretation (“A preferential loss of small, thin-shelled taxa in surface samples would produce misleading paleocommunity results”). In standard procedures we should focus on the whole material available (including the smallest taxa) to get the complete record of palaeoenvironmental conditions. “Smaller specimens would greatly increase sample size”, but do we know how the lack of them affect the comparisons presented in the paper? Are they still valid? This should be clarified in the text.

The second general problem is a lack of comparisons with other sites, deposits, environments etc. I see that this is a plan for future works, but I think that it should be clarified earlier than in conclusions (in methods or in the beginning of the discussion section). Maybe there are some literature data to compare with, that could be included in the paper?!

Specific comments:

Geological background - I suggest to add figure with lithological logs in studied outcrops with indication of the sampling points

L 49 – add “by” after influenced

L 111, 112, 116, 125, 129, 134 – should be Fig 1A and Fig 1B instead of 1a and 1b

L 244 – should be “multidimensional” instead of “multideminsional”

L 289 – remove double “who” in the caption of Table 1. Clarify NA abbreviation.

L 303-304 – Table 2: paired T-test, whereas t-test in the text; be consistent

L 324 – delete at in “letters at after”

L 406 – delete “is”

L 430 – remove dates from the citations

L 532 – start with a new paragraph

Fig. 1 – Check the scale and length of the histograms. They don’t match the values cited in text (lines 113-116: slab/section in paleoecology articles 11/22 – on the figure 25, other 18/32 – almost 40 etc.).

Fig. 2 – Show larger geographical context for international audience. Show the location on the map of the USA, or at least on its larger, more characteristic part. Put some coordinates or grid and more geographical/administrative names.

Fig. 5 – I suggest to provide more distinct differences in colors (darker vs. lighter). I’m not sure they will be distinguishable in final version.

Fig. 5 and 6 – If 1 is the oldest and 3 is the youngest, maybe its better to change the axes. Put histograms in stratigraphic order on axis y and relative abundance on axis x. If appropriate change also Fig. 8.

Reviewer #2: The paper is interesting and I would be happy to see it eventually published once the necessary corrections are introduced. I see several problems with this paper listed below.

Choice of references: It seems that the authors have chosen only US-based literature while there is much information on this and related topics in European papers. For example Cassian Formation studies (Nützel Kaim 2014, Hausmann et al. 2015) the differences in bulk sediment vs. surface sampling strategy are discussed with different conclusions than yours.

Strategy of sampling: One major flaw I see is 2 mm cut-off in bulk sediment samples. It potentially removes several small-sized taxa. Also changes considerably rank abundance. I personally use 0.375 mm mesh size to keep in the sample nearly all taxonomically identifiable mollusks. It is also worth mentioning that bulk sediment samples are advisable for silicified and phosphatized fossils (see e.g., Halamski et al. 2015 and Dzik 1994 respectively). Bulk sampling is also a natural choice for fine grained siliciclastics (black clays) where the majority of fossils is small-sized and well preserved while the rock itself is easy to disintegrate (e.g. Kaim 2001, 2011, 2012, Kaim Sztajner 2012 among others)

Contamination: Another source of major bias is possible contamination from overlying beds. Couldn’t it be overcome? You mentioned this possibility yourself but, really, comparison of contaminated (surface) vs. pristine (bulk) samples has a little sense in this respect.

Surface covered: You mentioned that surface collection comes from approx. 25-100 m2 area while bulk samples are collected from 0.1 m x 0.1 m = 0.01 m2 area that is an order of magnitude smaller. No wonder that you receive unequal sample sizes. Also by means of diversity (not all microenviroments occurring in 100 m2 will be represented by 0.01 m2 subsample. You should collect the bulk sample all over this 100 m2 area in order to get good representation.

Subsampling: Why no rarefaction performed. It would nicely address the issue of diversity change after subsampling. Also why no rank abundance given?

Finally I would recommend using bulk sediment vs. surface sampling instead in-place vs. surface. The latter sounds cumbersome, isn’t surface sampling also in-place?

References:

Dzik, J. 1994. Evolution of 'small shelly fossils' assemblages of the early Paleozoic. Acta Palaeontologica Polonica 38, 3, 247-313.

Halamski, A.T., Bitner, M.A., Kaim, A., Kolar-Jurkovšek, T., and Jurkovšek, B. 2015. Unusual brachiopod fauna from the Middle Triassic algal meadows of Mt. Svilaja (Outer Dinarides, Croatia). Journal of Paleontology 89(4): 553–575.

Hausmann, I.M. Nützel, A. 2015: Diversity and palaeoecology of a highly diverse Late Triassic marine biota from the Cassian Formation of north Italy. Lethaia 48: 235–255.

Kaim, A. 2001. Faunal dynamics of juvenile gastropods and associated organisms across the Valanginian transgression-regression cycle in central Poland. Cretaceous Research 22, 3, 333-351.

Kaim, A. 2011. Non-actualistic wood-fall associations from Middle Jurassic of Poland. Lethaia 34, 1, 109-124.

Kaim, A. 2012. Faunal dynamics of gastropods in the Bathonian (Middle Jurassic) ore-bearing clays at Gnaszyn, Kraków-Silesia Homocline, Poland. Acta Geologica Polonica, 62 (3), 367–380.

Kaim, A. and Sztajner, P. 2012. Faunal dynamics of bivalves and scaphopods in the Bathonian (Middle Jurassic) ore-bearing clays at Gnaszyn, Kraków-Silesia Homocline, Poland. Acta Geologica Polonica, 62 (3), 381–395.

Nützel, A. and Kaim, A. 2014. Diversity, palaeoecology and systematics of a marine fossil assemblage from the Late Triassic Cassian Formation at Settsass Scharte, N Italy. Paläontologische Zeitschrift 88 (4): 405-431.

Reviewer #3: The manuscript is well-written and easy to follow. The infuence of sampling on diversity is very relevant for our community. The methods used appropriate. I did find some minor things i would like to see resolved before publication (see also annotated pdf).

These mainly concern some additional references concerning influence of lithology on preservation/collection or a references summarizing diversity in the Finis Shale other than your own as well as some lithologies/localities (e.g., Liberation Lagerstätten) might be more prone to use the herein suggested methods.

Klug, C., Samankassou, E., Pohle, A., De Baets, K., Franchi, F., Korn, D. 2018. Oases of biodiversity: Early Devonian palaeoecology at Hamar Laghdad, Morocco. Neues Jahrbuch für Geologie und Paläontologie-Abhandlungen, 290(1-2), 9-48.

Lobza, V., Schieber, J. Merlynd, N. 1994. The Influence of Sea Level Changes and Possible Pycnocline Shifts on Benthic Communities in the Finis Shale (Virgilian) Near Jacksboro, North-Central Texas. Canadian Society of Petroleum Geologists Memoir, 17, 927-947.

Roden, V. J., Hausmann, I. M., Nützel, A., Seuss, B., Reich, M., Urlichs, M., ... Kiessling, W. 2019. Fossil liberation: a model to explain high biodiversity in the Triassic Cassian Formation. Palaeontology.

You share all data but it is hard to get an idea about preservation and determination of fossils within seeing at least some pictures of the main taxa - at least in the supplementary material. Some of these could also be integrated in the figures (e.g., pictures of the main brachiopod species integrated in figure 6).

In some cases, the NMDS labels (without points) are also hard to read (e.g., overlapping in Figs. 3 and 4). Some additional multivariate tests (e.g., Permanova as you did for the others) supporting the NMDS results for the subsampling might also be appropriate for the sake of consistency or would these not be appropriate in this case?

These and additional points are also listed in the annotated pdf.

6. PLOS authors have the option to publish the peer review history of their article (what does this mean?). If published, this will include your full peer review and any attached files.

Reviewer #1: No

Reviewer #2: Yes: Andrzej Kaim

Reviewer #3: Yes: Kenneth De Baets

---

## [Author Response · Author response to Decision Letter 0]

4 Dec 2019

See attached document that contains responses to each of the reviewers' comments.

---

## [Decision Letter · Decision Letter 1]

14 Jan 2020

PONE-D-19-25701R1

The influence of collection method on paleoecological datasets: In-place versus surface-collected fossil samples in the Pennsylvanian Finis Shale, Texas, USA

PLOS ONE

Dear Dr. Forcino,

Thank you for submitting your manuscript to PLOS ONE. After careful consideration, we feel that it has merit but does not fully meet PLOS ONE’s publication criteria as it currently stands. Therefore, we invite you to submit a revised version of the manuscript that addresses the points raised during the review process.

We would appreciate receiving your revised manuscript by Feb 28 2020 11:59PM. To enhance the reproducibility of your results, we recommend that if applicable you deposit your laboratory protocols in protocols.io, where a protocol can be assigned its own identifier (DOI) such that it can be cited independently in the future. For instructions see: http://journals.plos.org/plosone/s/submission-guidelines#loc-laboratory-protocols

We look forward to receiving your revised manuscript.

Kind regards,

Mikołaj K. Zapalski, Ph. D., D. Sc.

Academic Editor

PLOS ONE

Additional Editor Comments (if provided):

Dear Dr. Forcino,

Thank you very much for submitting the revsed version of the manuscript. Before the final decision I would like to ask you to consider a small development of conclusions - in the present version they are very general, while your manuscript brings large amount of quantitative data. Therefore I would suggest you to include in the conclusions and abstract some quantitative comparisons between bulk- and surface sampling.

Yours sincerely,

Mikołaj Zapalski

Reviewers' comments:

Reviewer's Responses to Questions

**Comments to the Author**

1. If the authors have adequately addressed your comments raised in a previous round of review and you feel that this manuscript is now acceptable for publication, you may indicate that here to bypass the “Comments to the Author” section, enter your conflict of interest statement in the “Confidential to Editor” section, and submit your "Accept" recommendation.

Reviewer #3: All comments have been addressed

2. Is the manuscript technically sound, and do the data support the conclusions?

Reviewer #3: Yes

3. Has the statistical analysis been performed appropriately and rigorously? 

Reviewer #3: Yes

4. Have the authors made all data underlying the findings in their manuscript fully available?

Reviewer #3: Yes

5. Is the manuscript presented in an intelligible fashion and written in standard English?

Reviewer #3: Yes

6. Review Comments to the Author

Reviewer #3: The manuscript is now even easier to follow. The authors addressed all my suggestions including adding a new figure of representative specimens. A better focus and a more homogenous background would make the latter even better. I have no additional comments.

7. PLOS authors have the option to publish the peer review history of their article (what does this mean?). If published, this will include your full peer review and any attached files.

Reviewer #3: Yes: Kenneth De Baets

---

## [Author Response · Author response to Decision Letter 1]

17 Jan 2020

“I would like to ask you to consider a small development of conclusions - in the present version they are very general, while your manuscript brings large amount of quantitative data. Therefore I would suggest you to include in the conclusions and abstract some quantitative comparisons between bulk- and surface sampling.”

We have added text to the abstract and conclusions that includes our statistical results.

---

## [Editor Report · Decision Letter 2]

28 Jan 2020

The influence of collection method on paleoecological datasets: In-place versus surface-collected fossil samples in the Pennsylvanian Finis Shale, Texas, USA

PONE-D-19-25701R2

Dear Dr. Forcino,

We are pleased to inform you that your manuscript has been judged scientifically suitable for publication and will be formally accepted for publication once it complies with all outstanding technical requirements.

With kind regards,

Mikołaj K. Zapalski, Ph. D., D. Sc.

Academic Editor

PLOS ONE

Additional Editor Comments (optional):

Dear Dr. Forcino,

You have integrated my remarks and I am happy to see this version of the manuscript. In its present form it is ready to be published.

Yours sincerely,

Mikołaj Zapalski
---

## [Editor Report · Acceptance letter]

29 Jan 2020

PONE-D-19-25701R2 

The influence of collection method on paleoecological datasets: In-place versus surface-collected fossil samples in the Pennsylvanian Finis Shale, Texas, USA 

Dear Dr. Forcino:

I am pleased to inform you that your manuscript has been deemed suitable for publication in PLOS ONE. Congratulations! Your manuscript is now with our production department. 

With kind regards,

on behalf of

Dr. Mikołaj K. Zapalski 

Academic Editor

PLOS ONE